# Baroclinic Instability of a Time-Dependent Zonal Shear Flow

**Chengzhen Guo and Jian Song \***

College of Science, Inner Mongolia University of Technology, Hohhot 010051, China; 18835512998@163.com

\* Correspondence: songjian@imut.edu.cn

**Abstract:** In the real atmosphere, the development of large-scale motion is often related to the baroclinic properties of the atmosphere. So, it is necessary to discuss the stability condition of baroclinic flow. It is advantageous to use a layered model to discuss baroclinic instability, not only to apply the potential vortex equation directly, but also to deal with shear of basic flow. The stability and oscillatory shear ability of Rossby waves are studied based on the two-layer Phillips model in the $\beta$ plane; then, we summarize the baroclinic instability of time-dependent zonal shear flows. The multiscale method is used to eliminate some terms of natural frequency oscillations of nonlinear operators in the third-order expansion, thus generating an equation about the amplitude of the lowest-order Rossby wave in the long-time variable. The large amplitude perturbation begins to decrease, which produces the desired behavior. After the amplitude decreases for some time, the amplitude of Rossby waves can still be found to oscillate periodically with the time variable.

**Keywords:** baroclinic instability; zonal basic flow; multiscale method

## 1. Introduction

Baroclinic instability theory is one of the mechanisms for the occurrence and development of large-scale atmospheric motion in middle and high latitudes. It is another important scientific advance in atmospheric dynamics after long wave theory. In most literature, researchers often idealize the basic flow as zonal westerly wind, which is quite different from the actual atmospheric state. In the real atmosphere, although zonal mean westerly winds are an important feature, in general, the atmospheric fundamental flow is not purely zonal and sometimes the meridional fundamental flow can be strong. Baroclinic instability of ocean currents can produce features of Gulf Stream vortices of orders of magnitude.

Pedlosky and Thomson [1] pointed out that in macroscopic background flows, the initial conditions for vortex development are described by the rapid growth of infinitesimal orthogonal mode disturbances, which can effectively obtain energy from large-scale background states due to their spatial and temporal structure. Drazin and Reid [2] reported that large-scale instability of mid-latitude westerly winds is an important problem in meteorology. This has been simulated in the laboratory using the instability of shear flows in a differentially heated rotating annulus. It is called baroclinic instability because it essentially depends on the difference between a constant density surface and a constant pressure surface in the fluid. Chen and Kamenkovich [3] outlined the effect of topography on baroclinic instability. The conditions of instability, the bounds of growth rates, and the phase velocities of unstable modes have been well-defined; even the theory of finite amplitudes has been greatly developed. Nevertheless, even small changes in the fundamental problem can produce substantial changes in the stability problem, and thus, give new characteristics to the dynamics of the waves and eddy current activity generated by such instability. In this paper, we study one form of this variation—the baroclinic instability of time-dependent zonal shear flow.

Flierl and Pedlosky [4] proposed the Baroclinic instability of subcritical fundamental stream functions with time through the Phillips [5] two-layer model on the $\beta$ plane. How-

ever, when the governing equation of perturbation is given in his thesis, the basic stream function does not consider the time dependence, which is not ideal. The classical theory of zonal flow instability in the $\beta$ plane has given a clear critical value for instability, but when the shear force is below the classical critical value, the time-dependent flow will exhibit instability. Parametric instability occurs when the frequency of the fundamental flow matches the multiple characteristic frequencies of other stable disturbances. This paper focuses on the nonlinear behavior of disturbance caused by parametric instability. We investigate the dynamics of the baroclinic instability of the Phillips [5] two-layer model in the $\beta$ plane and consider appropriate parametric values so that the basic state of the model is well below the critical value of the model instability. In the context of the shallow-water theory of disks, Umurhan [6] analyzed the dynamic normal mode response of thin annular disks with two strong local potential vorticity gradients and proved that baroclinic instability is feasible for astrophysical disks and has the characteristics of mixed baroclinic type. Moon et al. [7] studied the planetary scale fluctuations in the large-scale atmosphere by considering the parameterization of the planet-scale baroclinicity on the synoptic-scale heat flux based on the vortex memory effect.

## 2. The Small *H* Limit Model

Take the two-layer Phillips model on the $\beta$ plane as reference. Setting channel width L, we consider zonal flow without horizontal shear. For the sake of simplicity, the thickness of each layer is assumed to be D when there is no movement. $\psi_n$ is the geostrophic stream function for each layer, where $n = 1$ represents the upper layer and $n = 2$ represents the lower layer. The nondimensional governing equations are [8]

$$\frac{\partial q_n}{\partial t} + J(\psi_n, q_n) + \beta \frac{\partial \Psi_n}{\partial x} = -\mu q_n, n = 1, 2. \tag{1}$$

where

$$q_n = \nabla^2 \psi_n + (-1)^n F(\psi_1 - \psi_2), \tag{2}$$

$$F = \frac{f_0^2 L^2}{g' D}, \tag{3}$$

$$\beta = \frac{\beta_{dim} L^2}{U_{scale}}. \tag{4}$$

Define the nondimensional parameter $F$ as the ratio of channel width to deformation radius, and $\beta$ is the ratio of planetary vorticity gradient to the characteristic value of relative vorticity gradient, all set to $O(1)$. J is the Jacobian operator. A dissipative mechanism with a rate constant $\mu$ is introduced on the right side of (1) as the deboost of potential vorticity. The basic flow of disturbance is $\phi(x, y, t)$, The total steamfunction is

$$\psi_n = (-1)^n y \frac{U_s}{2} + \phi_n. \tag{5}$$

The governing equations for the perturbations are

$$[\nabla^2 \phi_n + (-1)^n F(\phi_1 - \phi_2)] + \frac{\partial \phi_n}{\partial x} [\beta - (-1)^n F U_s] + J[\phi_n, \nabla^2 \phi_n + (-1)^n F(\phi_1 - \phi_2)]$$
$$-(-1)^n F y (\frac{\partial U_s}{\partial t} + \mu U_s) = 0. \tag{6}$$

In the absence of dissipation, if $U_s$ is time-independent, then the critical value required for perturbation growth will be $\frac{\beta}{F}$[1]. Due to the potential vorticity deboost in (6), the critical value will be slightly higher than $O(\mu)$. We are concerned with the case where the

fundamental shear changes over time and is always less than this critical value. Consider the basic form of shearing as follows:

$$U_s = \frac{\beta}{F}(G + H\cos(\omega t)).$$ (7)

when $G + H < 1$ the shear force at every moment is below the critical value. The baroclinic model and baroclinic model of disturbed field are used to reformulate the problem. The following definitions describe the baroclinic and baroclinic modes

$$\psi_t = \frac{\phi_1 + \phi_2}{2},$$ (8)

$$\psi_c = \frac{\phi_1 - \phi_2}{2}.$$ (9)

It can be obtained, respectively, according to (6)

$$(\frac{\partial}{\partial t} + \mu)(\nabla^2\phi_c - 2F\phi_c + FU_sy) + \frac{U_s}{2}\frac{\partial}{\partial x}(\nabla^2\phi_t + 2F\phi_t) + \beta\frac{\partial\phi_c}{\partial x} + J(\phi_t, \nabla^2\phi_c - 2F\phi_c) + J(\phi_c, \nabla^2\phi_t) = 0,$$ (10)

$$(\frac{\partial}{\partial t} + \mu)\nabla^2\phi_t + \frac{U_s}{2}\frac{\partial}{\partial x}\nabla^2\phi_c + \beta\frac{\partial\phi_t}{\partial x} + J(\phi_t, \nabla^2\phi_t) + J(\phi_c, \nabla^2\phi_c) = 0.$$ (11)

Consider the case where there is no average shear force when $G = 0$. For $H << 1$, the flow at every moment is much less than the critical value $\frac{\beta}{F}$[1]. According to Flierl and Pedlosky's research [4], we set $\frac{\beta}{F} = 1$. This defines the scaling speed of the shear, which is the Rossby long wave velocity $\frac{\beta_{dim}(g'D)}{f_0^2}$. Assume that the amplitude expansion of each flow function is

$$\phi_{c,t} = a[\phi_{c,t}^{(0)} + a\phi_{c,t}^{(1)} + ...],$$ (12)

The slow time scale is $T = Ht$, Assuming that $a = O(H^{\frac{1}{2}})$ and $\mu = O(H)$, we define $\tilde{\mu} = \frac{\mu}{H}$. A geostrophic flow function can be both a function of $T$ and $t$. Therefore, the time derivatives in (10) and (11) can be converted to

$$\frac{\partial}{\partial t} \Rightarrow \frac{\partial}{\partial t} + H\frac{\partial}{\partial T},$$ (13)

The amplitude expansion solution for the lowest-order baroclinic and baroclinic Rossby modes is

$$\phi_t^{(0)} = B_t(t, T)e^{ikx}e^{-ikt\frac{c_c + c_t}{2}}\sin ly + *,$$ (14)

$$\phi_c^{(0)} = B_c(t, T)e^{ikx}e^{-ikt\frac{c_c + c_t}{2}}\sin ly + *.$$ (15)

where an asterisk denotes complex conjugation, and

$$c_t = -\frac{\beta}{K^2},$$ (16)

$$c_c = -\frac{\beta}{K^2 + 2F},$$ (17)

$$k^2 = k^2 + l^2,$$ (18)

$$l = m\pi.$$ (19)

where the asterisk indicates complex conjugate. In order to conform the situation that there is no normal current at the channel boundary, the wave number $y$ needs to be an

integer multiple of $\pi$. In the following, we will choose $m = 1$, which is the lowest and most unstable mode. Then, the governing equation of amplitude variable has the following form

$$
\left.\begin{array}{l}
\frac{\partial B_c}{\partial t} - ik\frac{c_t - c_c}{2}B_c = 0, \\
\frac{\partial B_t}{\partial t} + ik\frac{c_t - c_c}{2}B_t = 0.
\end{array}\right\} \Rightarrow B_t = B_{t0}(T)e^{i\sigma t}, B_c = B_{c0}(T)e^{i\sigma t} \tag{20}
$$

where $\sigma = k\frac{c_t - c_c}{2}$. Frequency $\sigma$ is the critical frequency that defines the parametric instability of oscillating shear resonance. In the first order of $a$, these waves propagate as free Rossby waves. As we will see, parametric instability occurs at a node of frequency $2\sigma$, where the amplitude is still an arbitrary function of the long time variable $T$. In the next order of $a$, the nonlinear term applies force only in (10), and the force has no connection with $x$. At this point,

$$
\phi_t^{(1)} = 0, \tag{21}
$$

and

$$
\phi_c^{(1)} = \Phi(y, t, T). \tag{22}
$$

$\Phi$ satisfies

$$
\frac{\partial}{\partial t}(\nabla^2\Phi - 2F\Phi) - 2iklF\sin 2ly(B_t B_c^* - B_c B_t^*) + Fy\frac{\partial}{\partial t}U_s = 0. \tag{23}
$$

The solution of $\Phi$ can be obtained as

$$
\Phi = P(t, T)\left(\frac{-lF}{2l^2 + F}\right)\left(\sin 2ly - \frac{2l}{\sqrt{2F}}\frac{\sinh\sqrt{2F}(y - \frac{1}{2})}{\cosh\sqrt{2F}}\right) + \frac{U_s}{2}\left(y - \frac{1}{\sqrt{2F}}\frac{\sinh\sqrt{2F}(y - \frac{1}{2})}{\cosh\sqrt{\frac{F}{2}}}\right) \tag{24}
$$

and

$$
P(x, y) = \frac{-k}{2\sigma}(B_{t0}B_{c0}^*e^{-2i\sigma t} + B_{t0}^*B_{c0}e^{2i\sigma t}). \tag{25}
$$

The solution can be obtained by using boundary conditions $\frac{\partial\Phi}{\partial y} = 0, y = 0, 1$ [8]. We corrected the oscillation frequency of the average flow to $2\sigma$, and no time average is included in the oscillation period. In the absence of a time average of the fundamental shear force, the shear flow does not exist in the $y$ direction. Then, we do not expect the thickness flux of baroclinic flow to be changed in the unstable wave, and can choose the direction of the integral flux that can produce the time-averaged flux correction. Next, we will examine the average flow correction in the presence of an average but subcritical base-state shear. When expanding to the next order of $a$, we can obtain the linear problem of higher-order correction of the geostrophic flow function. The time-related questions are obtained in the form of (20), The terms on both sides of the equation oscillate at the same natural frequency. These terms will produce more and more terms in the expansion of $a$; so, we need to eliminate these terms to make them invalid in other ways. Under the requirement of slow scale time $T$, the amplitude equations of Rossby waves are obtained as follows:

$$
(\frac{\partial}{\partial t} + \widetilde{\mu})B_{t0} - \frac{\beta}{2F}\frac{ik}{2}R_1 B_{c0} + \frac{ik}{2\sigma}N_1 B_{t0}|B_{c0}|^2 = 0, \tag{26}
$$

$$
(\frac{\partial}{\partial t} + \widetilde{\mu})B_{c0} + \frac{\beta}{2F}\frac{ik}{2}R_2 B_{t0} - \frac{ik}{2\sigma}N_2 B_{c0}|B_{t0}|^2 = 0. \tag{27}
$$

where

$$
R_1 = \frac{1}{K^2}[-2K^2 - (2F - K^2)\frac{4l^2}{4l^2 + 2F}\frac{\tanh\sqrt{\frac{F}{2}}}{\sqrt{\frac{F}{2}}}], \tag{28}
$$

$$
R_2 = \frac{K^2}{K^2 + 2F}\frac{4l^2}{4l^2 + 2F}\frac{\tanh\sqrt{\frac{F}{2}}}{\sqrt{\frac{F}{2}}}. \tag{29}
$$

The fundamental shear must oscillate at the frequency $\omega = 2\sigma = k(c_t - c_c)$ in order to suppress terms that may resonate with the linear operator in (20). If it were not for this frequency, the second term of (26) and (27) from the interaction between oscillating shear and the lowest-order Rossby waves would not have occurred. Nonlinear systems (26) and (27) control the amplitude of the Rossby waves of (14) and (15); so, (24) and (25) are also used to describe the correction for the mean zonal flow. Where

$$N_1 = \frac{4l^2 kF}{HK^2(4l^2 + 2F)}\Big[\frac{K^2}{2} - 2l^2 + \frac{4l^2(2F + K^2)}{4l^2 + 2F}\frac{\tanh\sqrt{\frac{F}{2}}}{\sqrt{\frac{F}{2}}}\Big]. \tag{30}$$

$$N_2 = \frac{4l^2 kF}{H(K^2 + 2F)(4l^2 + 2F)}\Big[\frac{-K^2}{2} + 2l^2 + F - \frac{4l^2 K^2}{4l^2 + 2F}\frac{\tanh\sqrt{\frac{F}{2}}}{\sqrt{\frac{F}{2}}}\Big]. \tag{31}$$

Firstly, the nonlinear terms in (26) and (27) are temporarily ignored, and the stability of Rossby waves and their capacity for change on oscillatory shear are discussed through these linear terms. If Rossby waves are unstable, this produces the expected behavior when small amplitude perturbations begin to grow. Suppose the form of the solution is $e^{\alpha T}$ or $e^{\alpha Ht}$. We obtain the following growth rate

$$\alpha H = \frac{k\beta}{4F}H\sqrt{R_1 R_2} - \mu. \tag{32}$$

Thus, oscillating shear is unstable when the amplitude exceeds the critical value below

$$H_{crit} = \frac{4F\mu}{\beta k}\sqrt{\frac{1}{R_1 R_2}}. \tag{33}$$

The short-wave cut-off of the shear is consistent with the standard stability problem. The shear flow frequency of $2\sigma$ is not required. As described below, the frequency range for which this parametric instability occurs is generally enlarged for larger $H$. After a period of exponential growth, the nonlinear terms in (26) and (27) can no longer be ignored. Nevertheless, nonlinear solutions of (26) and (27) can still be solved. Their solutions are unstable and oscillate periodically with slow time scale $T$. The nonlinear solution has the following form:

$$B_{t0} = B'_{to}e^{i\varpi T} \tag{34}$$

and

$$B_{c0} = B'_{co}e^{i\varpi T}. \tag{35}$$

Multiplied by the complex conjugate of the positive and baroclinic amplitudes, the real and imaginary parts of (26) and (27) can be expressed as follows:

$$|B_{t0}|^2 = \frac{R_1}{R_2}|B_{c0}|^2 \tag{36}$$

and

$$\varpi = \frac{k}{4\sigma}|B_{c0}|^2\Big(N_1 - \frac{R_1}{R_2}N_2\Big). \tag{37}$$

After substituting (36) and (37) into (26) and (27), the amplitude of the oscillation is obtained

$$|B_{c0}|^2 = \frac{4\sigma}{k}\left[\frac{\sqrt{\frac{\beta^2 k^2 R_1 R_2}{16F^2} + \widetilde{\mu}^2}}{|3N_1 - \frac{R_1}{R_2}N_2|}\right]. \tag{38}$$

According to (10) and (14), finite amplitude solutions can exist only when linear solutions are unstable. In addition, the predicted final state from (32) and (38) contains a frequency

shift proportional to the square of the equilibrium amplitude. According to the prediction of (36)–(38), the oscillation amplitude of the solution gradually decreases, and finally, the amplitude balances to a stable value. Figures 1 and 2 show the real and imaginary parts of the positive and baroclinic amplitudes, respectively.

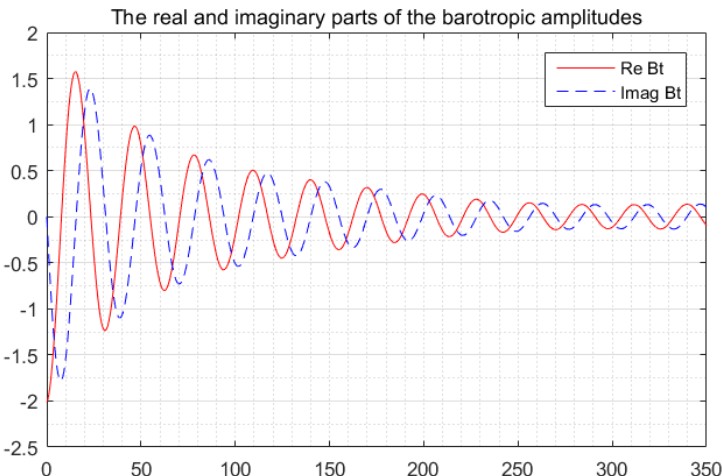

**Figure 1.** Asymptotically small $H$ solutions developed on long time scales: $T = Ht$. Real and imaginary parts of barotropic Rossby wave amplitude. The results in each case match the analytical solutions of (36)–(38). The calculations are performed for $\mu = 0.015, F = \beta = 20, H = 0.05$, and $k = l = \pi$.

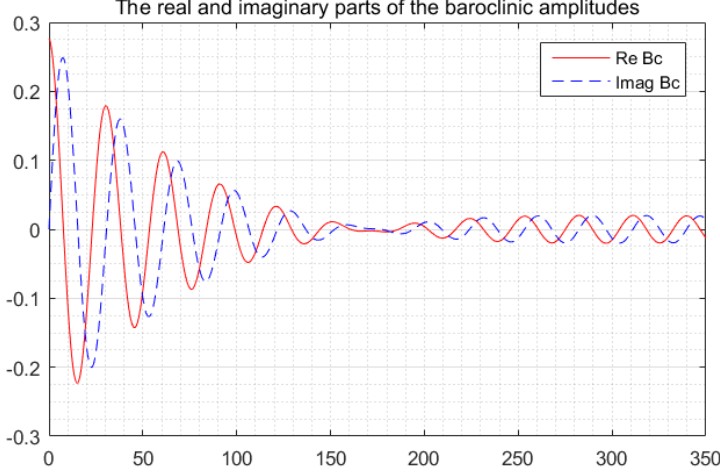

**Figure 2.** Asymptotically small $H$ solutions developed on long time scales: $T = Ht$. Real and imaginary parts of baroclinic Rossby wave amplitude. The results in each case match the analytical solutions of (36)–(38). The calculations are performed for $\mu = 0.015, F = \beta = 20, H = 0.05$, and $k = l = \pi$.

Figure 3 shows the evolution of the absolute value of the amplitudes.

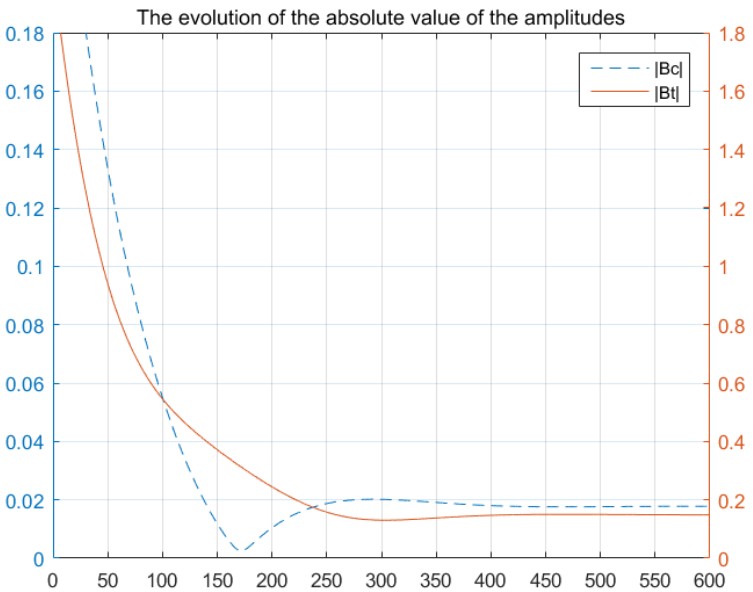

**Figure 3.** The calculations are performed for $\mu = 0.015, F = \beta = 20, H = 0.05$, and $k = l = \pi$.

## 3. Discussion

With the passage of time, the amplitude of barotropic Rossby wave oscillates and the amplitude of oscillation gradually decreases with the increase in time, and finally tends to be steady. With the passing of time, the amplitude of baroclinic Rossby wave oscillates and becomes smaller and smaller with the increase in time. When the oscillation reaches a critical point, the amplitude will be close to zero, and then gradually increases to a steady value.

In the two-layer Phillips model, according to the classical critical value $\frac{\beta}{F}$ of baroclinic instability given by Pedlosky and Thomson [1], they suggest in their study that zonal flows with shear values much less than $\frac{\beta}{F}$ are unstable when the flow is time-dependent. Thus, under the condition of $H \ll 1$, in Equation (7), the basic form of shear flow is expanded to obtain the term $\frac{\beta}{F}H$. Therefore, although the shear value is much lower than the traditional stability critical value at every instant, we still think that the classical steady flow can become very unstable. Unstable waves with high energy content strongly alter the mean flow. The results show that the time-dependence of zonal basic flow greatly affects baroclinic instability. This has obvious implications for the problem of vortex effects in ocean and atmospheric flow patterns.

**Author Contributions:** C.G. wrote the first draft of the paper and contributed images. J.S. contributed to the theoretical derivation. All authors have read and agreed to the published version of the manuscript.

**Funding:** The work is supported by the National Natural Science Foundation of China (Grant No.41765004) and Development plan of young scientific and technological talents in Colleges and Universities in Inner Mongolia (Grant No. NMGIRT2208).

**Institutional Review Board Statement:** Not applicable.

**Informed Consent Statement:** Not applicable.

**Data Availability Statement:** The data that support the findings of this study are in the attachment.

**Acknowledgments:** Thanks to Jian Song for his theoretical guidance and revision suggestions.

**Conflicts of Interest:** The authors declare no conflict of interest.

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
