# Peer review of "Baroclinic Instability of a Time-Dependent Zonal Shear Flow"

_atmosphere, doi:10.3390/atmos13071058_

Round 1

Reviewer 1 Report

1. Please state more clearly the improvement of your method compared to former studies in section 2.

2. As in the discussion(line 173-175), zonal time-dependent flows with shear value much less than the classical critical value(beta/F)  are unstable, can you give more specific value of the beta/F in time dependent flow? How low is this value compared with that in Pedlosky[8]?

Also, is there any observation support for the results in this paper?

Please add these discussions in section 4.

Reviewer 2 Report

This is my second reviewer report on this article. In my previous review report, I recommended the authors to make some improvements to the manuscript. I also submitted a list of specific comments, which the authors responded to in general, but not in full. The term "flow functions" has not been changed to "stream functions" throughout the manuscript. However, my main critical comment about the need for a more detailed comparison with the results of Pedlosky and Thomson (2003) went largely unanswered. The authors simply deleted the sentence “And in this study, the shear value is much weaker than that in Pedlosky and Thomson’s paper” from Section 4 Discussion. Reading the manuscript, the reader may get the mistaken feeling that these authors are the first to study the parametric baroclinic instability of a time-dependent vertical shear flow in the framework of the two-layer Phillips model. However, this is not true, and at least Pedlosky and Thomson (2003) have done this before. Therefore, I am so insistent that the authors discuss in more detail the issue of difference and novelty of their work compared to Pedlosky and Thomson (2003).

Round 2

Reviewer 1 Report

This version is improved.

This manuscript is a resubmission of an earlier submission. The following is a list of the peer review reports and author responses from that submission.

Round 1

Reviewer 1 Report

This article studied the stability condition of baroclinic flow. It is an old topic with many related studies. The authors failed to summarize clearly the background of this topic. Some of the citation of the literature is even problematic. For example, the paper by Pedlosky and Thomson is not published on 2020 in JAS. It should be the one published on 2003 on  Journal of Fluid Mechanics. The work by Farrell and Ioannou (1999), to my knowledge, they mainly stated that the jet is contribute to the generation of cyclones, not as the authors said in line 32-33 "the mean flow weakens the instability". Another paper ChangHeng and Kamenkovich (2013) should be Chen and Kamenkovich (2013).

There are many related works in the literature. The authors should have made a thorough investigation before writing this paper. There're two references listed in below.

Baroclinic instability and large-scale wave propagation in a planetary-scale atmosphere.  Moon et al. 2022 Q J R Meteorol Soc. 2022;148:809–825. DOI: 10.1002/qj.4232  

Potential vorticity dynamics in the framework of disk shallow-water theory.

Umurhan 2012 A&A 543, A124 (2012) DOI: 10.1051/0004-6361/201218803

Reviewer 2 Report

The authors present a study of the time dependent zonal flow and their de-stabilization through the baroclinic instability. The motivation and the aim of the article is not clear, the article must be re-written so that it can be understandable to the scientific community. The results section also needs to be explained thoroughly, the methods and what the results indicate. 

Reviewer 3 Report

The authors study the linear and nonlinear stages of development of parametrically unstable perturbations superimposed on a vertical shear flow with time-dependent amplitude, within the framework of the classical two-layer Phillips model. To the best of my knowledge, this paper is an immediate follow-up to the article by J. Pedlosky and J. Thomson entitled “Baroclinic instability of time-dependent currents” and written on essentially the same topic, but it is not clear to me why that article is referred to as being published in J. Atmos. Sci. 2020, 142–149. I have not been able to find such an article in the indicated journal, but instead I know that an article by the same authors and with exactly the same title has been published in J. Fluid Mech. (2003), Vol. 490, pp. 189–215. DOI: 10.1017/S0022112003005007. I personally think the last reference is more correct, but the authors have to double check it.

I have not followed in detail all the derivations of the manuscript, but I consider them done correctly and the results obtained as reasonable and useful. My main critical comment is about the need for a more detailed comparison with the results of Pedlosky and Thomson (2003). The authors write in Section 4 Discussion “And in this study, the shear value is much weaker than that in Pedlosky and Thomson’s paper”. But what exactly does it mean? Does it mean that the systematic part of the shear becomes zero, G=0, whereas in Pedlosky and Thomson (2003) it does not? Overall, how new is the current work compared to Pedlosky and Thomson (2003)?

Several specific comments.

  1. “(as in low exponential circulation)” on line 20 (page 1) is incomprehensible.
  2. “flow functions” on line 43 (page 2) and elsewhere has to be changed to “stream functions”.
  3. “Arametric” on line 48 and “parameter” on line 50 (page 2) should be changed to “parametric”.
  4. Equation (4) on page 2 needs to be improved.
  5. What is “pressure flow” on line 67 (page 2)?
  6. Line 141 (page 5): it is better to change “truncation” to “cut-off”.
  7. Line 145 (page 5) and almost everywhere else: “stable” should be changed to “steady”; see also Pedlosky and Thomson (2003).

In general, I am a bit concerned about the English writing style of the authors. It has to be improved. Ideally, Pedlosky and Thomson (2003) could be followed in this regard.